# Bridging Non Co-occurrence with Unlabeled In-the-wild Data for Incremental Object Detection

**Na Dong**[1,2*]     **Yongqiang Zhang**[2]     **Mingli Ding**[2]     **Gim Hee Lee**[1]

[1]Department of Computer Science, National University of Singapore
[2]School of Instrument Science and Engineering, Harbin Institute of Technology
`{dongna1994, zhangyongqiang, dingml}@hit.edu.cn`     `gimhee.lee@comp.nus.edu.sg`

## Abstract

Deep networks have shown remarkable results in the task of object detection. However, their performance suffers critical drops when they are subsequently trained on novel classes without any sample from the base classes originally used to train the model. This phenomenon is known as catastrophic forgetting. Recently, several incremental learning methods are proposed to mitigate catastrophic forgetting for object detection. Despite the effectiveness, these methods require co-occurrence of the unlabeled base classes in the training data of the novel classes. This requirement is impractical in many real-world settings since the base classes do not necessarily co-occur with the novel classes. In view of this limitation, we consider a more practical setting of complete absence of co-occurrence of the base and novel classes for the object detection task. We propose the use of unlabeled in-the-wild data to bridge the non co-occurrence caused by the missing base classes during the training of additional novel classes. To this end, we introduce a blind sampling strategy based on the responses of the base-class model and pre-trained novel-class model to select a smaller relevant dataset from the large in-the-wild dataset for incremental learning. We then design a dual-teacher distillation framework to transfer the knowledge distilled from the base- and novel-class teacher models to the student model using the sampled in-the-wild data. Experimental results on the PASCAL VOC and MS COCO datasets show that our proposed method significantly outperforms other state-of-the-art class-incremental object detection methods when there is no co-occurrence between the base and novel classes during training. Our source code is available at `https://github.com/dongnana777/Bridging-Non-Co-occurrence`.

## 1 Introduction

Deep learning have shown remarkable performance in a wide variety of tasks, and even surpass human experts in numerous tasks. However, humans are still better than machines in continually acquiring, fine-tuning and transferring knowledge throughout their lifetime. This is because deep networks suffer from catastrophic forgetting [18, 21], *i.e.* a phenomenon that causes a deep network to forget previously acquired knowledge on the old base classes when trained on new novel classes. As a result, this causes the performance of the deep networks to drop significantly on the base classes. In the task of image-based object detection, object detectors are trained on large-scale datasets and then deployed on real-world applications [9, 11, 8, 24, 5, 15, 23, 17, 16, 1, 2, 39, 33, 34, 38, 36, 37, 35, 32]. As these applications are often carried out in dynamic environments where novel object classes are continually presented, the ability for the deep networks to learn novel object classes without forgetting the base object classes becomes an imperative requirement. A naive approach to achieve this endeavor

---

*Work fully done while first author is a visiting PhD student at the National University of Singapore.

35th Conference on Neural Information Processing Systems (NeurIPS 2021).

Table 1: Co-occurrence statistics of the base classes in the novel datasets from PASCAL VOC under various splits of "base + novel" classes.

| setting | number of objects | aero | bicycle | bird | boat | bottle | bus | car | cat | chair | cow | diningtable | dog | horse | motorbike | person | pottedplant | sheep | sofa | train | tv |
|---|---|---|---|---|---|---|---|---|---|---|---|---|---|---|---|---|---|---|---|---|---|
| 19+1 | base classes | 2 | 14 | 0 | 0 | 108 | 0 | 8 | 28 | 440 | 0 | 38 | 28 | 0 | 0 | 344 | 116 | 0 | 148 | 0 | - |
| | novel classes | - | - | - | - | - | - | - | - | - | - | - | - | - | - | - | - | - | - | - | 734 |
| 15+5 | base classes | 4 | 24 | 8 | 6 | 260 | 2 | 140 | 112 | 1094 | 4 | 164 | 156 | 20 | 8 | 1594 | - | - | - | - | - |
| | novel classes | - | - | - | - | - | - | - | - | - | - | - | - | - | - | - | 1250 | 706 | 850 | 656 | 734 |
| 10+10 | base classes | 74 | 666 | 158 | 188 | 890 | 256 | 1398 | 180 | 2492 | 246 | - | - | - | - | - | - | - | - | - | - |
| | novel classes | - | - | - | - | - | - | - | - | - | - | 620 | 1076 | 812 | 780 | 10894 | 1250 | 706 | 850 | 656 | 734 |

is to retain all training data for the base classes and train the deep network concurrently with the base and novel training data from scratch. However, pragmatic issues such as privacy can limit the accessibility to the base class dataset previously used to train the base model.

Recently, several incremental learning methods [40, 27, 4, 10, 20, 31] are proposed to overcome catastrophic forgetting in the object detection task. Despite the impressive performance, these methods rely on the co-occurrence of unlabeled base classes in the training data of the novel classes. Due to the fact that the base and novel class datasets are obtained from the same dataset to simulate incremental learning, the base classes inevitably co-occur in the background of the novel class training data. Table 1 shows the co-occurrence statistics of the base classes in the novel datasets from PASCAL VOC under several commonly used experiment settings in the existing works. For example, class 1-19 are chosen as the base classes and class 20 ("tv") as the novel class. We can see from the table that the "areo", "bicycle", "bottle", *etc.* , objects in the base classes co-occur in the sample images of the novel "tv" object class. Despite the absence of ground truth labels of the base classes on the novel training data, knowledge of base classes can be replayed and distilled into the novel model. However, this reliance on the co-occurrence of unlabeled base classes in the training data of novel classes severely limits the practicality of most incremental learning approaches. This is due to the inherent non co-occurrence of the base and novel classes in most real-world data.

In this paper, we propose the use of the abundance in-the-wild data to bridge the non co-occurrence of base classes in the training data of the novel classes. To this end, we first introduce a blind sampling strategy to select relevant data from the in-the-wild data that contains large amounts of irrelevant images with neither the base nor novel class information. Specifically, images with high probability response from the given base model and the novel model trained with the novel class training data are selected in the blind sampling step. We further design a dual-teacher distillation framework where the images selected from the blind sampling strategy are used to distill knowledge from the base and novel teacher models to the student model. Particularly, our dual-teacher distillation framework consists of: 1) A class remodeling step that remodels the irrelevant classes as background class in the base and novel model, respectively. This ensures the class probabilities of the disjoint set of classes in the base and novel models can be compared appropriately to the student model. 2) Image-level distillation with region of interest (RoI) masks from the pseudo ground truth of the bounding boxes obtained in the blind sampling step. These RoI masks are used to mask out the irrelevant regions of the feature maps in the distillation loss. 3) Instance-level distillation with the response heatmaps of the object detectors. This response heatmaps is essential in transferring both positive (high response regions) and negative (low response regions) information from the base and novel teacher models to the student model.

Our main contributions are as follows:

1. We tackle a more challenging and realistic scenario of incremental learning for object detection, where there is no co-occurrence of the base classes in the training data of the novel classes. This contrasts with previous approaches whose success depend largely on such co-occurrences.

2. We propose a blind sampling strategy to effectively select useful data from large amounts of unlabeled in-the-wild data. We also design a dual-teacher distillation framework which utilizes the sampled data to distill knowledge from the base and novel teacher models to the student model.

3. Extensive experiments conducted on two standard object detection datasets (*i.e.* MS COCO and PASCAL VOC) demonstrate the significant performance improvement of our approach over existing state-of-the-art.

## 2 Related Works

### 2.1 Object Detection

Existing deep object detection models fall generally into two categories: 1) One-stage detectors and 2) Two-stage detectors. One-stage detectors such as YOLO [23] directly performed object classification and bounding box regression on the feature maps. SSD [17] uses feature pyramid with different anchor sizes to cover the possible object scales. RetinaNet [16] proposes the focal loss to mitigate the imbalanced positive and negative examples. Two-stage detectors such as R-CNN [9] apply a deep neural network to extract features from proposals generated by selective search [28]. Fast R-CNN [8] improves the speed and performance utilizing a differentiable RoI Pooling. Faster R-CNN [24] introduces Region Proposal Network(RPN) to generate proposals. FPN [15] builds a top-down architecture with lateral connections to extract features across multiple layers. Typically, both one-stage and two-stage object detectors require large amounts of training images per class and need to train the detectors over many training epochs. Unfortunately, it is unlikely that large amounts of data for the old classes are present in the new training data. Therefore, the extension of the capability of detectors to novel categories with no access to the original training data is imperative.

### 2.2 Class-incremental Learning

Most works on class-incremental learning focus on the image classification problems, and can be roughly divided into two major families: 1) regularization approaches and 2) rehearsal approaches. In the regularization approaches, learning capacity is assumed to be fixed and incremental learning is performed so that the change in parameters is controlled or reduced. Kirkpatrick *et. al* [13] propose the elastic weight consolidation (EWC) method in which $\Omega_i$ is calculated as diagonal approximation of the empirical Fisher Information Matrix. The second type of regularization-based approaches is based on knowledge distillation [14, 6]. Li *et. al* [14] propose to use Learning without Forgetting (LwF) to keep the representations of base data from drifting too much while learning the novel tasks. In the rehearsal approaches, the models strengthen memories learned in the past through replaying the past information periodically. They usually keep a small number of exemplars [22, 29, 3], or generate synthetic images [26, 19] or features [12, 30] to achieve this purpose. The rehearsal method for class-incremental learning is first proposed in iCaRL [22], and has been applied in majority of the class-incremental learning works.

### 2.3 Class-incremental Object Detection

Class-incremental object detection are relatively less explored than its image classification counterpart. All the proposed methods follow the regularization-based approach, in which knowledge distillation is used to address the catastrophic forgetting issue. Shmelkov *et. al* [27] proposes a Fast R-CNN-based class-incremental object detection model to address the catastrophic forgetting problem, where EdgeBoxes [41] is used to produce bounding box proposals. However, the proposals generation stage introduces an immense computational cost. Sub-optimal performance for the base and novel classes are obtained since a sub-optimal two-stage object detection model is used. Zhou *et. al* [40] proposes an incremental version of Faster R-CNN that distills selected anchors and proposals with the Pseudo-Positive-Aware Sampling strategy. However, it fails to improve the benchmark [27] on the standard evaluation criteria. The class-incremental object detection methods mentioned above are impractical in the real-world since the key success factor is the utilization of co-occurred unlabeled base objects with novel objects in the novel training data. Zhang *et. al* [31] presents a class-incremental learning paradigm called Deep Model Consolidation (DMC) for single-shot object detection architectures, which combines the base model and the novel model leveraging external unlabeled auxiliary data. In this work, we develop a novel mechanism for the task of class-incremental object detection based on the Faster R-CNN framework. Our framework is a dual-teacher that distill knowledge from the teacher base and novel models to the student model using sampled unlabeled in-the-wild data, without requiring the co-occurrence of base and novel object classes.

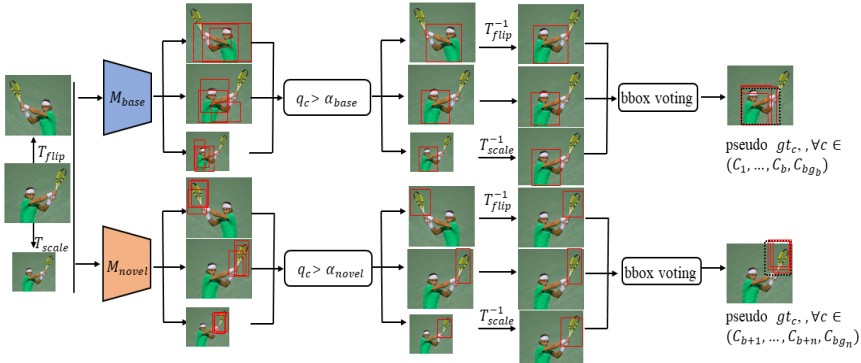

Figure 1: Overview of our proposed blind sampling strategy. Refer to the text for more details.

## 3   Our Approach

### 3.1   Preliminaries

Let $(x, y) \in \mathcal{D}$ denotes a dataset $\mathcal{D}$ which contains images $x$ and their corresponding object bounding box labels $y$. We further denote the training dataset of the base classes and the novel classes as $\mathcal{D}_{\text{base}}$ and $\mathcal{D}_{\text{novel}}$, respectively. Following the definition of class-incremental learning, we only have access to the novel class data $\mathcal{D}_{\text{novel}}$, where $y_{\text{novel}} \in \{C_{b+1}, \ldots, C_{b+n}, C_{\text{bg}_n}\}$. The base class data $\mathcal{D}_{\text{base}}$, where $y_{\text{base}} \in \{C_1, \ldots, C_b, C_{\text{bg}_b}\}$ are no longer accessible. $C_{\text{bg}_n}$ and $C_{\text{bg}_b}$ are the background class of the novel and base data, respectively. Unlike other existing incremental learning methods for object detection where the base classes can still occur as unlabeled background classes in the novel training data, we enforce the more realistic strict non co-occurrence $\{C_{b+1}, \ldots, C_{b+n}, C_{\text{bg}_n}\} \cup \{C_1, \ldots, C_b, C_{\text{bg}_b}\} = \emptyset$ in our training data. We further assume that a large quantity of unlabeled in-the-wild data is accessible, from which we sample $\mathcal{D}_{\text{unlabel}}$ using our blind sampling strategy. The base model $\mathcal{M}_{\text{base}}(\mathcal{D}_{\text{base}}; \theta_{\text{base}})$ is an object detector trained on the base class data, where $\theta_{\text{base}}$ denotes the learnable parameters. Our goal is to train an object detection model $\mathcal{M}_{\text{stud}}(\mathcal{D}_{\text{unlabel}}; \theta_{\text{stud}})$ to detect the novel classes $\{C_{b+1}, \ldots, C_{b+n}, C_{\text{bg}_n}\}$ without catastrophic forgetting the base classes $\{C_1, \ldots, C_b, C_{\text{bg}_b}\}$. We use Faster R-CNN [24] as our object detector.

### 3.2   Blind Sampling Strategy

Although large amounts of unlabeled in-the-wild data are easily obtainable, most of them are not useful for alleviating the catastrophic forgetting problem. Naive training on these unlabeled in-the-wild data increases training time and can be detrimental to the network performance. We propose to circumvent this problem by sampling useful data from the large amounts of unlabeled in-the-wild data to build the sampled unlabeled dataset $\mathcal{D}_{\text{unlabel}}$. Furthermore, it is important to note that it is not necessary for the in-the-wild data to contain any object in the base and novel classes. We postulate that false positives from the in-the-wild data with non-overlapping object classes from the base and novel classes can also serve as useful training data in our dual-teacher framework. This is based on the intuition that the features maps and response heatmaps of false positives closely resemble the positive samples.

As shown in Fig. 1, we use do the blind sampling with the base $\mathcal{M}_{\text{base}}(\mathcal{D}_{\text{base}}; \theta_{\text{base}})$ and novel $\mathcal{M}_{\text{novel}}(\mathcal{D}_{\text{novel}}; \theta_{\text{novel}})$ models that are pre-trained on the base $\mathcal{D}_{\text{base}}$ and novel $\mathcal{D}_{\text{novel}}$ training data, respectively. Specifically, we first feed the input image into both $\mathcal{M}_{\text{base}}$ and $\mathcal{M}_{\text{novel}}$. The image is selected if the object class probability $q_c, \forall c \in \{C_1, \ldots, C_b, C_{\text{bg}_b}\}$ in the base model $\mathcal{M}_{\text{base}}$ or $\forall$

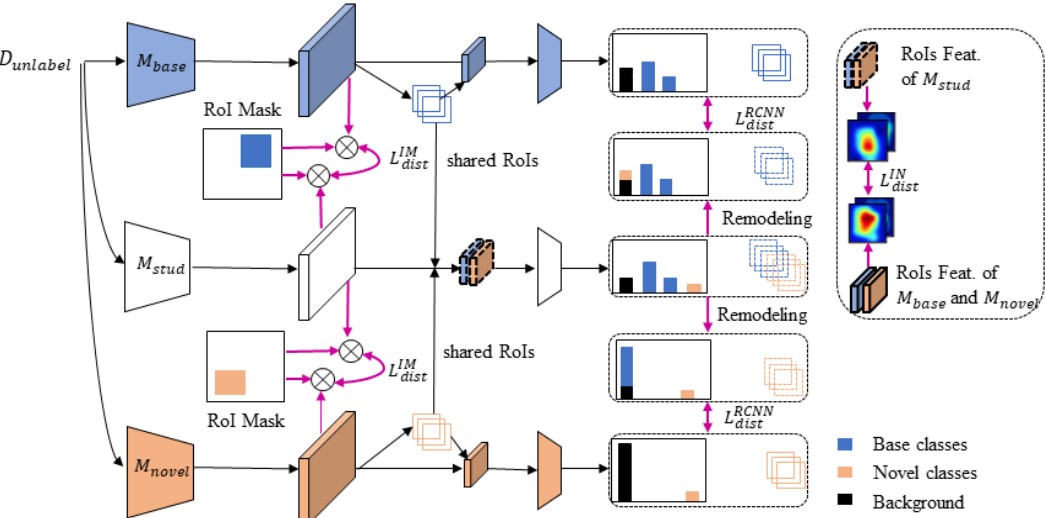

Figure 2: Overview of our dual-teacher distillation framework for object detection incremental learning. Blue, orange and black branches denote the dual-teacher base and novel models, and the student model, respectively. Dash pink lines denote the knowledge distillation process with our proposed loss functions $\mathcal{L}_{dist}^{RCNN}$, $\mathcal{L}_{dist}^{IM}$ and $\mathcal{L}_{dist}^{IN}$. Best viewed in color.

$c \in \{C_{b+1}, \ldots, C_{b+n}, C_{bg_n}\}$ in the novel model $\mathcal{M}_{novel}$ is greater than a pre-defined threshold. We use two different thresholds: 1) $\alpha_{base}$ for $\mathcal{M}_{base}$, and 2) $\alpha_{novel}$ for $\mathcal{M}_{novel}$. We further enhance the accuracy and precision of the blind sampling strategy by a random transformation consistency check. To this end, we augment the input image by applying random scaling $T_{scale}$ and horizontal flipping $T_{flip}$. The original image, the randomly scaled and the randomly flipped images are then fed into the object detection network. We then apply the inverse scaling $T_{scale}^{-1}$ and horizontal flipping $T_{flip}^{-1}$ on the respective outputs. The input image is selected into $\mathcal{D}_{unlabel}$ only if the predicted outputs are consistent over the random transformations. We improve the precision of the estimated bounding boxes by bounding box voting [7], where the ensemble results of the regression bounding boxes from the augmented images are used as the pseudo ground truths in our subsequent dual-teacher distillation framework.

## 3.3 Dual-Teacher Distillation

As illustrated in Fig. 2, we use the base $\mathcal{M}_{base}$ and novel $\mathcal{M}_{novel}$ models as the dual teachers in our dual-teacher knowledge distillation framework. The objective is to train a student model $\mathcal{M}_{stud}$ that inherits the ability to detect all the foreground object classes in the base and novel classes, i.e. $\{C_1, \ldots, C_b, C_{b+1}, \ldots, C_{b+n}, C_{bg_{stud}}\}$. We denote the background class of the student model as $C_{bg_{stud}}$. All $\mathcal{M}_{base}$, $\mathcal{M}_{novel}$ and $\mathcal{M}_{stud}$ are the same Faster R-CNN object detection network.

**Remodeling prediction outputs.** For knowledge distillation on the object detection head of Faster R-CNN, we first select the informative foreground region of interests (RoIs) proposals from the 256 candidates generated by the region proposal network (RPN) of the two teacher models, respectively. Subsequently, the combination of the selected RoIs from the base $\mathcal{M}_{base}$ and novel $\mathcal{M}_{novel}$ models are shared to the student model $\mathcal{M}_{stud}$. These selected foreground RoIs are passed through the R-CNN module of the student model to compute the prediction outputs $(q_c^{stud}, r_c^{stud})$, where $q$ is predicted probability and $r$ is coordinates of the predicted bounding box. The output foreground RoIs of the two teacher models $(q_c^{base}, r_c^{base})$ and $(q_c^{novel}, r_c^{novel})$ serve as the targets. However, the student model is a $b + n$-class object detector, while the base model and the novel model are a $b$-class and a $n$-class object detector, respectively. This means that the output object class probabilities and the bounding boxes between the teacher and student models cannot be compared directly.

To alleviate this problem, we propose the remodel the prediction outputs of the student model in accordance to the targeted teacher model. For the distillation of the base model, we remodel the

non-overlapping classes from the student model with the base model into the background class of the base model: $\tilde{q}_c^{\text{stud,base}} = \{(q_{\text{bg}}^{\text{stud}} + q_{c_{b+1}}^{\text{stud}} + \cdots + q_{c_{b+n}}^{\text{stud}}), q_{c_1}^{\text{stud}}, q_{c_2}^{\text{stud}}, \ldots, q_{c_b}^{\text{stud}}\}$. Similarly for the distillation of the novel model, we remodel the non-overlapping classes from the student model with the novel model into the background class of the novel model, *i.e.* $\tilde{q}_c^{\text{stud, novel}} = \{(q_{\text{bg}}^{\text{stud}} + q_{c_1}^{\text{stud}} + q_{c_2}^{\text{stud}} + \cdots + q_{c_b}^{\text{stud}}), q_{c_{b+1}}^{\text{stud}}, \ldots, q_{c_{b+n}}^{\text{stud}}\}$. We do the same remodeling on the regression head of the object detectors. The regression outputs $r_c^{\text{stud}}$ of the student model are remodeled into two parts $\tilde{r}_c^{\text{stud,base}}$ and $\tilde{r}_c^{\text{stud,novel}}$ according to the number of shared RoIs with the base and novel models, respectively.

Finally, the output class probabilities and bounding boxes of the student and teacher models are be directly compared in the R-CNN distillation loss function given by:

$$
\begin{aligned}
\mathcal{L}_{\text{dist}}^{\text{RCNN}} = {} & \mathcal{L}_{\text{kl\_div}}(\log(\tilde{q}_c^{\text{stud,base}}), q_c^{\text{base}}) + \lambda \mathcal{L}_{\text{smooth}_{L_1}}(\tilde{r}_c^{\text{stud,base}}, r_c^{\text{base}}) \\
& + \mathcal{L}_{\text{kl\_div}}(\log(\tilde{q}_c^{\text{stud,novel}}), q_c^{\text{novel}}) + \lambda \mathcal{L}_{\text{smooth}_{L_1}}(\tilde{r}_c^{\text{stud,novel}}, r_c^{\text{novel}}),
\end{aligned}
\tag{1}
$$

where $\mathcal{L}_{\text{kl\_div}}$ is the KL-divergence loss between the class probabilities of the student and teacher models. Following [8], $\mathcal{L}_{\text{smooth}_{L_1}}$ is a robust L1 loss between the bounding box parameters of the student and teacher models. $\lambda$ is a hyperparameter to balance the KL-divergence and Smooth-L1 losses. Intuitively, $\mathcal{L}_{\text{dist}}^{\text{RCNN}}$ supervises the student model $\mathcal{M}_{\text{stud}}$ to produce outputs that are close to the base $\mathcal{M}_{\text{base}}$ and novel $\mathcal{M}_{\text{novel}}$ teacher models.

**Image-level distillation with RoI masks.** [25] shows that intermediate-level supervisions from the hidden layers of the teacher model can guide the student model towards better generalization than supervision on only the output predictions. However, the approach cannot be naively applied to our incremental learning setting. A direct knowledge distillation on the full feature maps of the dual teachers causes conflicts and thus hurts the overall performance. Specifically, the base model teacher would wrongly instruct the student model to suppress the novel classes as the background class, and vice versa. To mitigate these conflicts, we use the pseudo ground truth bounding boxes of the foreground classes from the blinding sampling step as RoI masks $Mask^{\text{base}}$ and $Mask^{\text{novel}}$. These RoI masks are applied to the feature map distillation loss to focus the attention on the regions of interest. Concurrently, these masks prevent the confusion of the background classes from the two teacher models. We write the image-level distillation loss with the RoI masks on the feature maps as:

$$
\mathcal{L}_{\text{dist}}^{\text{IM}} = \frac{1}{2N^{\text{base}}} \sum_{i=1}^{W} \sum_{j=1}^{H} \sum_{k=1}^{C} Mask_{ij}^{\text{base}} \left\| F_{ijk}^{\text{stud}} - F_{ijk}^{\text{base}} \right\|^2 + \frac{1}{2N^{novel}} \sum_{i=1}^{W} \sum_{j=1}^{H} \sum_{k=1}^{C} Mask_{ij}^{\text{novel}} \left\| F_{ijk}^{\text{stud}} - F_{ijk}^{\text{novel}} \right\|^2,
\tag{2}
$$

where $N^{\text{base}} = \sum_{i=1}^{W} \sum_{j=1}^{H} Mask_{ij}^{\text{base}}$, $N^{\text{novel}} = \sum_{i=1}^{W} \sum_{j=1}^{H} Mask_{ij}^{\text{novel}}$. $F^{\text{base}}$, $F^{\text{novel}}$ and $F^{\text{stud}}$ denote the feature of the teacher base and novel models, and the student model, respectively. $W$, $H$ and $C$ are the width, height and channels of the feature map.

**Instance-level distillation with heatmaps.** Since the image-level distillation can lead to domination at instance-level, we introduce an instance-level distillation to balanced attentions on all instances. We define the instance-level distillation loss with the response heatmaps of the object detection models. Each location on the heatmap and its response represent the degree of influence on the model prediction outputs from a pixel on the input image. Let us denote the features of the object proposals generated by the base, novel and student models as $f^{\text{base}}$, $f^{\text{novel}}$ and $f^{\text{stud}}$, respectively. The heatmaps of the base, novel and student models are then given by channel-wise average pooling and element-wise Sigmoid activation $S(.)$, *i.e.* $\mathcal{H}_{ij}^{\text{base}} = S(\frac{1}{C} \sum_{k=1}^{C} f_{ijk}^{\text{base}})$, $\mathcal{H}_{ij}^{\text{novel}} = S(\frac{1}{C} \sum_{k=1}^{C} f_{ijk}^{\text{novel}})$, and $\mathcal{H}_{ij}^{\text{stud}} = S(\frac{1}{C} \sum_{k=1}^{C} f_{ijk}^{\text{stud}})$ over the spatial locations $i = 1, \ldots, W$ and $j = 1, \ldots, H$. To enforce consistency between the student model and the two teacher models at instance level, we design the instance-level distillation loss $\mathcal{L}_{\text{dist}}^{\text{IN}}$ as the mean squared error (MSE) between the student heatmap $\mathcal{H}^{\text{stud}}$ and the union of the heatmaps $\mathcal{H}^{\text{base}} \cup \mathcal{H}^{\text{novel}}$ from the two teacher models, *i.e.*

$$
\mathcal{L}_{\text{dist}}^{\text{IN}} = \mathcal{L}_{\text{mse}}(\mathcal{H}^{\text{base}} \cup \mathcal{H}^{\text{novel}}, \mathcal{H}^{\text{stud}}).
\tag{3}
$$

**Overall loss.** The overall loss $\mathcal{L}_{\text{total}}$ to train the student model incrementally on the sampled unlabeled in-the-wild dataset $\mathcal{D}_{\text{unlabel}}$ given by:

$$
\mathcal{L}_{\text{total}} = \mathcal{L}^{\text{RCNN}} + \mathcal{L}^{\text{RPN}} + \alpha_1 \mathcal{L}_{\text{dist}}^{\text{RCNN}} + \alpha_2 \mathcal{L}_{\text{dist}}^{\text{IM}} + \alpha_3 \mathcal{L}_{\text{dist}}^{\text{IN}},
\tag{4}
$$

where $\mathcal{L}^{\text{RCNN}}$ and $\mathcal{L}^{\text{RPN}}$ are the loss terms for the R-CNN and RPN module of the two-stage detector Faster R-CNN, supervised by the pseudo ground truths from the blind sampling step. $\alpha_1, \alpha_2$, and $\alpha_3$ are the hyperparameters to weigh the loss terms.

# 4 Experiments

## 4.1 Experimental Setup

**Datasets and metrics.** Following [27], we evaluate our proposed method for class-incremental object detection on the PASCAL VOC 2007 and MS COCO 2014 datasets. PASCAL VOC 2007 consists of about 5K training and validation images and 5K test images over 20 object categories. Models are trained on the trainval set and tested on the test set. MS COCO 2014 contains objects from 80 different categories with 83K images in the training set and 41K images in the validation set. We train models on the training set and evaluate models on the first 5K images of the validation set. In the test stage, the mean average precision (mAP) is used as the evaluation metrics. We report the COCO style (mAP [0.5 : 0.95]) detection accuracy for MS COCO dataset, and PASCAL style (mAP [0.5]) accuracy for PASCAL VOC dataset. Additional average precision and recall across scales are also reported, which is in line with standard evaluation protocol of MS COCO. To evaluate the compared methods under the setting with a large amount of in-the-wild unlabeled data for the PASCAL VOC and MS COCO target datasets, we take the MS COCO and Open Images datasets as the unlabeled data. We also run the experiments under a more strict setting that excludes all MS COCO images that contain any object instance of 20 PASCAL VOC categories to avoid any potential advantage from the classes of PASCAL VOC appearing in MS COCO. We denote this setting as "$w/o\ category$".

**Implementation details.** We use ResNet-50 with frozen batch normalization layers as the backbone network. The training methodology is the same as standard Faster R-CNN. We use stochastic gradient descent with Nesterov momentum to train the models in all experiments. The initial learning rate is set to 1e-3 and subsequently reduced by 0.1 after every 5 epochs for the previous model and the current model. Each model is trained for 20 epochs for both PASCAL VOC and MS COCO datasets. The training is carried out on 1 RTX 2080Ti GPU, and the batch size is set to 1.

## 4.2 Comparison of Methods

Table 2: Results of "19+1" on VOC $test$ set. "1-19" and "20" ("tv") are base and novel classes. "base | novel | all" is mAPs for base, novel and all classes. Row 1-3 are baselines without incremental learning.

| Class | Method | mAP(%) ($base \mid novel \mid all$) |
|---|---|---|
| 1-20 (w/o co-occur) | Ren [24] | 73.1 \| 55.4 \| 72.3 |
| 1-19 (w/o co-occur) | Ren [24] | 73.4 \| − \| − |
| 20 (w/o co-occur) | Ren [24] | − \| 47.4 \| − |
| (1-19) + (20) (w/o co-occur) | Shmelkov [27] | 62.6 \| 39.2 \| 61.4 |
| | Ours (w category) | **73.3** \| **50.7** \| **72.2** |
| | Ours (w/o category) | **71.5** \| **46.1** \| **70.2** |
| (1-19) + (20) (w co-occur) | Shmelkov [27] | 68.5 \| 62.7 \| 68.3 |
| | Zhou [40] | 70.5 \| 53.0 \| 69.6 |
| | Ours (w category) | **73.5** \| **65.8** \| **73.1** |

**Addition of one class.** Table 2 shows the results of one addition class incremental learning. We take the first 19 classes in alphabetical order from PASCAL VOC as the base classes $C_1, ..., C_b$ and the remaining class is used as the novel class $C_{b+1}$. In addition to the disjoint base and novel classes following the definition of class-incremental learning, we also exclude images that have co-occurrence of any base and novel objects. We report the mean average precision (mAP) of the base, novel and all classes, which we denote as $base \mid novel \mid all$. The first three rows show the evaluation results of the "1-20", "1-19", "20" baselines without using incremental learning. Furthermore, we compare to the state-of-the-art incremental object detection method [27] on training data with ("$w\ co-occur$") and without ("$w/o\ co-occur$") co-occurrence of the base and novel classes. In the absence of co-occurrence, we can see that [27] suffers a drop in the performance of the novel class from 62.7% to 39.2% mAP, and base class from 68.5% to 62.6% mAP. It should also be noted that the mAPs of [27] are significantly lower than the baselines without incremental learning. In contrast, our approach without co-occurrence and with class overlap in the in-the-wild data ("$w\ category$") achieves mAPs of 73.3% | 50.7% | 72.2% (base | novel | all). Our result is almost on par with the results of the baseline training with all classes "1-20" without incremental learning at 73.1% | 55.4% | 72.3%. Additionally, we can also see that our method still achieves high mAP of 71.5% | 46.1% | 70.2% even when

there is no class overlap in the in-the-wild data ("$w/o\ category$"). This supports our postulation that false positives sampled from the in-the-wild data can also benefit our dual-teacher incremental learning framework. Interestingly, we can also run our dual-teacher distillation framework on the "$w\ co-occur$" data. It can be seen that we significantly outperform [27] and [40].

Table 3: Results of "15+5" on VOC $test$ set. "1-15" and "16-20" are the base and novel classes.

| Class | Method | mAP(%) ($base \mid novel \mid all$) | | |
|-------|--------|------|---|---|
| 1-20 (w/o co-occur) | Ren [24] | 74.4 | 62.7 | 71.7 |
| 1-15 (w/o co-occur) | Ren [24] | 72.0 | – | – |
| 16-20 (w/o co-occur) | Ren [24] | – | 48.6 | – |
| (1-15) + (16-20) (w/o co-occur) | Shmelkov [27] | 67.2 | 46.1 | 62.0 |
| | Ours (w category) | **70.5** | **49.4** | **65.3** |
| | Ours (w/o category) | **70.7** | **48.5** | **65.1** |
| Class (1-15) + (16-20) (w co-occur) | Shmelkov [27] | 68.4 | 58.4 | 65.9 |
| | Ours (w category) | **72.7** | **58.4** | **69.1** |

Table 4: Results of "10+10" on VOC $test$ set. "1-10" and "11-20" are the base and novel classes.

| Class | Method | mAP(%) ($base \mid novel \mid all$) | | |
|-------|--------|------|---|---|
| 1-20 (w/o co-occur) | Ren [24] | 66.5 | 69.0 | 67.7 |
| 1-10 (w/o co-occur) | Ren [24] | 57.8 | – | – |
| 11-20 (w/o co-occur) | Ren [24] | – | 63.2 | – |
| (1-10) + (11-20) (w/o co-occur) | Shmelkov [27] | **58.5** | 49.1 | 53.8 |
| | Ours (w category) | 57.6 | **62.2** | **59.9** |
| | Ours (w/o category) | **58.5** | **62.3** | **60.4** |
| (1-10) + (11-20) (w co-occur) | Shmelkov [27] | 63.2 | 63.1 | 63.1 |
| | Zhou [40] | 63.5 | 60.0 | 61.8 |
| | Ours (w category) | **69.2** | **68.3** | **68.7** |

**Addition of a group of classes.** Table 3 shows the results on 5 novel classes. We can see that our proposed approach "$w/o\ co-occur$" and "$w\ \&\ w/o\ category$" achieves mAPs that are close to the baseline (without incremental learning) on the base (Ours "$w\ \&\ w/o\ category$": 70.7% & 70.5% vs. "1-15": 72%) and novel (Ours "$w\ \&\ w/o\ category$": 48.5% & 49.4% vs. "16-20": 48.6%) classes. Furthermore, we achieve higher performances compared to [27] (Ours: 70.7 | 48.5 | 65.1 vs. [27]: 67.2 | 46.1 | 62.0) when trained without co-occurrence of the base and novel classes. Table 4 shows the results on 10 novel classes. We can see that our methods "$w/o\ co-occur$" and "$w\ \&\ w/o\ category$" achieve performances that are almost on par with the baselines without incremental learning on the base (Ours "$w\ \&\ w/o\ category$": 57.6% & 58.5% vs "1-10": 57.8%) and novel (Ours "$w\ \&\ w/o\ category$": 62.2% & 62.3% vs. "11-20": 63.2%) classes. Moreover, our method significantly outperforms [27] on the novel classes when trained without co-occurrence (Ours "$w\ category$": 62.2%; Ours "$w/o\ category$": 62.3% vs. [27]: 49.1%). In Tables 3 and 4, we also show that our method outperforms [27] and [40] under the "$w\ co-occur$" setting. These results show the effectiveness of our proposed approach when a group of novel classes are added.

Table 5: Results of "10+5+5" on VOC $test$ set. "1-10" are the base classes, and "11-15" and "16-20" are the two groups of sequentially added novel classes.

| Class | Method | mAP(%) ($base \mid novel \mid all$) | | |
|-------|--------|------|---|---|
| 1-20 (w/o co-occur) | Ren [24] | 66.6 | 67.3 | 66.7 |
| 1-10 (w/o co-occur) | Ren [24] | 57.8 | – | – |
| 11-15 (w/o co-occur) | Ren [24] | – | 62.4 | – |
| 16-20 (w/o co-occur) | Ren [24] | – | 48.6 | – |
| (1-10)+ (11-15) (w/o co-occur) | Shmelkov [27] | **59.8** | 52.4 | 57.3 |
| | Ours (w category) | 57.0 | **62.7** | **58.9** |
| | Ours (w/o category) | 57.3 | **61.7** | **58.8** |
| (1-10)+ (11-15)+ (16-20) (w/o co-occur) | Shmelkov [27] | **59.0** | 47.3 | 53.1 |
| | Ours (w category) | 56.7 | **55.1** | **55.8** |
| | Ours (w/o category) | 56.9 | **53.9** | **55.4** |
| (1-10)+ (11-15)+ (16-20) (w co-occur) | Zhou [40] | 60.3 | 53.1 | 56.7 |
| | Ours (w category) | **68.1** | **64.8** | **66.5** |

**Addition of classes sequentially.** To prove that our method is also efficient for sequential incremental learning, we train the model by sequentially adding new groups of novel classes. In our experiments, we use "1-10" as the base classes, and "11-15" and "16-20" are two groups of novel classes to be sequentially added. Tables 5 shows the results of sequential incremental learning. We can see that our method shows on par performance with [27] under "$w/o\ co-occur$" on the base classes when the novel class group of "11-15" (Ours "$w\ \&\ w/o\ category$": 57.0% & 57.3% vs. [27]: 59.8%) and "11-15"+"16-20" (Ours $w\ \&\ w/o\ category$: 56.7% & 56.9% vs. [27]: 59.0%) are added. Under the same settings, our method significantly outperforms [27] on the sequential addition of the two groups of novel classes "11-15" (Ours "$w\ \&\ w/o\ category$": 62.7% & 61.7% vs. [27]: 52.4%) and "11-15"+"16-20" (Ours "$w\ \&\ w/o\ category$": 55.1% & 53.9% vs. [27]: 47.3%). We also outperform [40] significantly under the "$w\ co-occur$" setting.

To demonstrate the generality of our method, we conduct an experiment using the validation set of Open Images as the unlabeled data. Specifically, we sample $1.5 \times 10^4$ images for $\mathcal{D}_{\text{unlabel}}$ from the validation set of Open Images with our blind sampling strategy in the "19+1" setting.

Table 7: Results of "40+40" on COCO $minival$ set. First 40 classes are the old classes, and the next 40 are the added classes.

| Class | Method | AP | AP50 | AP75 | APS | APM | APL |
|---|---|---|---|---|---|---|---|
| 1-80 (w/o co-occur) | Ren [24] | 27.7 | 45.8 | 29.4 | 10.8 | 30.9 | 42.5 |
| (1-40) + (40-80) (w/o co-occur) | Ours | **22.5** | **40.9** | **23.0** | **8.3** | **25.9** | **34.6** |
| (1-40) + (40-80) (w co-occur) | Shmelkov [27] | 21.3 | 37.4 | - | - | - | - |
| | Zhou [40] | 22.7 | 36.8 | - | - | - | - |
| | Ours | **23.7** | **42.5** | **24.3** | **8.6** | **26.6** | **37.5** |

Table 6: Results of "19+1" on VOC $test$ set. "1-19" and "20" are the base and added novel classe(s).

| Class | Method | mAP(%) ($base \mid novel \mid all$) |
|---|---|---|
| 1-20 (w/o co-occur) | Ren [24] | 73.1 \| 55.4 \| 72.3 |
| 1-19 (w/o co-occur) | Ren [24] | 73.4 \| $-$ \| $-$ |
| 16-20 (w/o co-occur) | Ren [24] | $-$ \| 47.4 \| $-$ |
| (1-19) + (20) (w/o co-occur) | Shmelkov [27] | 62.6 \| 39.2 \| 61.4 |
| | Ours | **71.3** \| **48.6** \| **70.1** |
| (1-19) + (20) (w co-occur) | Shmelkov [27] | 68.5 \| 62.7 \| 68.3 |
| | Zhou [40] | 70.5 \| 53.0 \| 69.6 |

The results are shown in Table 6. We can see that a remarkable performance improvement of 8.7% and 9.4% mAP is achieved by our method compared to [27] for the base and novel classes, respectively. In addition, we significantly outperform [27] by a large margin of 8.7% mAP on all classes. These results demonstrate the superior ability of our proposed method for class-incremental object detection.

Table 7 shows the results of further experiments on COCO with the test set of Open Images as the unlabeled data. In particular, we split the 80 classes into a "40+40" setup. For fair comparison, we report results on the minival dataset following [27]. Our method achieves a better performance of 22.5% mAP without co-occurrence of base objects and novel objects compared to [27] with the unfair advantage of co-occurrence. Furthermore, we can see that our method significantly outperforms [27] and [40] under the "$w\ co-occur$" setting.

### 4.3 Ablation Studies

Figure 3: Results of "19+1" on VOC $test$ set with varying amount of unlabeled data.

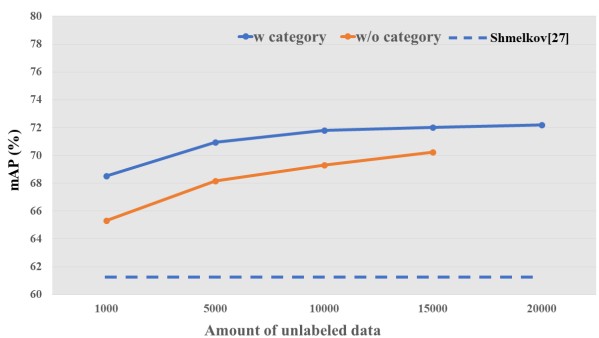

**Effect of the amount of unlabeled data.** Fig. 3 illustrates the effect of the amount of unlabeled data used for training. For "19+1", we sample $10^3, 5 \times 10^3, 10^4, 1.5 \times 10^4, 2 \times 10^4$ images using our blind sampling strategy for the "$w\ category$" setting. We sample $10^3, 5 \times 10^3, 10^4, 1.5 \times 10^4$ images for the "$w/o\ category$" due to the limited amount of "$w/o\ category$" data in unlabeled MS COCO dataset. We report the mean average precision over all steps. Overall, our method can even outperform the previous state-of-the-art by only using $10^3$ unlabeled "$w/o\ category$" images.

**Effect of the main configurations.** Table 8 shows the ablation studies to understand the effectiveness of each component in our framework. The ablated components include: 1) R-CNN distillation; 2) Image-level distillation; 3) Instance-level distillation. The first row is the result of only the standard Faster R-CNN loss, trained with pseudo ground-truth data generated by the two teacher models on the unlabeled in-the-wild dataset. We can see that this setting gives the lowest performance, which is evident on the importance of our three distillation losses. The subsequent inclusions of the three respective distillation losses in Row 2-5 show improvements over the standard Faster-RCNN baseline. These results further demonstrate the effectiveness of

Table 8: Ablation studies for the setting of "19+1" on VOC 2007 *test* set.

| Blind sampling strategy | $\mathcal{L}^{\text{RCNN}} + \mathcal{L}^{\text{RPN}}$ | $\mathcal{L}^{\text{RCNN}}_{\text{dist}}$ | $\mathcal{L}^{\text{IM}}_{\text{dist}}$ | $\mathcal{L}^{\text{IN}}_{\text{dist}}$ | mAP(%) (*base* \| *novel* \| *all*) |
|:---:|:---:|:---:|:---:|:---:|:---:|
| ✓ | ✓ | | | | 64.4 \| 35.3 \| 62.9 |
| ✓ | ✓ | ✓ | | | 68.0 \| 44.9 \| 66.9 |
| ✓ | ✓ | | ✓ | | 68.5 \| 39.6 \| 67.1 |
| ✓ | ✓ | | | ✓ | 67.6 \| 39.2 \| 66.2 |
| ✓ | ✓ | ✓ | ✓ | ✓ | 70.0 \| 44.3 \| 68.7 |
| ✓ | ✓ | ✓ | ✓ | ✓ | **71.5** \| **46.1** \| **70.2** |

the remodeling prediction outputs ($\mathcal{L}^{\text{RCNN}}_{\text{dist}}$), RoI masks ($\mathcal{L}^{\text{IM}}_{\text{dist}}$) and heatmaps ($\mathcal{L}^{\text{IN}}_{\text{dist}}$) on our method. Finally, the last row shows the best performance with our proposed blind sampling strategy and distillation losses.

## 5 Conclusion

In this paper, we present a novel class-incremental object detection for a more challenging and realistic scenario when there is no co-occurrence of base and novel object classes in images of the novel training dataset. This contrasts with other existing methods whose key success factor is the co-occurrence. We propose the use of unlabeled in-the-wild data to bridge the non co-occurrence caused by the missing base classes during the training of additional novel classes. A blind sampling strategy is proposed to select a smaller set of relevant data for incremental learning. We then design a dual-teacher knowledge distillation framework with three levels of distillation losses to transfer knowledge from the base- and novel-class teacher models to the student model using the sampled in-the-wild data. Extensive experimental results on benchmark datasets show the effectiveness of our proposed method.

## 6 Acknowledgement

The first author is funded by a scholarship from the China Scholarship Council (CSC). This research is supported in part by the National Research Foundation, Singapore under its AI Singapore Program (AISG Award No: AISG2-RP-2020-016) and the Tier 2 grant MOET2EP20120-0011 from the Singapore Ministry of Education.

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
