# OpenReview forum: "Bridging Non Co-occurrence with Unlabeled In-the-wild Data for Incremental Object Detection"
_NeurIPS.cc/2021/Conference — NeurIPS 2021 Poster_

### Official Review · Reviewer_m17V · 2021-07-12

**Rating:** 6
**Confidence:** 3

**Summary:**

The paper proposes a more realistic setting for class-incremental object detection, removing the need for co-occurrence if base and novel object classes in images in the training dataset. They propose a slightly complex method to address this scenario by assuming co-occurrence of the base and novel classes in supplementary, unlabeled data and distilling a model which learns from that data along with the training set using three levels of distillation losses and separate teacher models for base classes and novel classes that are distilled into a single, joint student model.

**Ethical Concerns:**

I don't see ethical considerations beyond the potential biases in the underlying datasets they use.

**Limitations And Societal Impact:**

They have not addressed this as far as I can tell. One possible limitation is the assumption that all classes of interest will occur in the unsupervised data, allowing them to avoid forgetting. In real-world scenarios or applications there may not be a source of unlabeled data containing examples of all classes to distill from.

**Main Review:**


This paper points out the co-occurrence in training data assumption underlying much of the recent work in incremental object detection. The assumption that novel classes will co-occur with examples of the base classes is still central to this work, the main difference as I understand it based on this paper is the access to larger unlabeled datasets to have a larger pool that is likely to provide examples of base and novel classes.  This point would be made stronger with some additional data analysis and visualization. In the novel class data subsets in question, how frequently do novel classes occur with no co-occurrence to base ones? How many base classes are not represented at all? I know these datasets are not fully labeled, but your best model’s predictions could be used as an estimate of the level of co-occurrence/non-occurrence and the number of examples of each class in the training data (required by the SOA methods cited in related work) vs the co-occurrence and number of examples seen in the unlabeled data.

The paper only compares to a single benchmark, which it outperforms across the board. However, if I am understanding correctly this benchmark does not get to see the significant amount of unsupervised data that their model does. How much of the performance gain is just through seeing more and more varied examples of the classes? Is it possible to disambiguate the performance gain from your methodology/proposed loss structure vs the performance gain from the unsupervised data? Or is the design of this methodology inseparable from the access to novel data to distill from?

Some minor grammatical errors throughout. Some additional editing for grammar is needed.


**Time Spent Reviewing:**

3

---

> ### Author Response · Authors · 2021-08-10
> **Our response to Reviewer m17V**
>
> **Q1: In the novel class data subsets, how frequently do novel classes occur with no co-occurrence to base ones?**
>
> **A:** The following table shows the co-occurrence frequency count of objects from the base and novel classes in the novel dataset from PASCAL VOC: [Number of objects on novel dataset](https://docs.google.com/presentation/d/1xE4oMZ_4wd9jLii-wQJxme3rqJUQQ6Xq/edit?usp=sharing&ouid=104968873642613184581&rtpof=true&sd=true). We can see that objects from the base classes still occur very frequently (even exceeding the frequency counts of novel objects in some cases) in the novel dataset, i.e. strong co-occurrences in all experimental settings.
>
>
> **Q2: You should disambiguate the performance gain from your methodology/proposed loss structure vs the performance gain from the unsupervised data.**
>
> **A:** We cannot measure our performance gain from the unsupervised data as our approach cannot work without the unsupervised data. Leveraging the unsupervised data to bridge the non-co-occurrence of base and novel classes for incremental object detection is our key contribution. Specifically, we sample useful in-the-wild data in an unsupervised manner, and these sampled data then serve as a link between the two teacher models and the student model.
>
>
> **Q3: Minor grammatical errors.**
>
> **A:** We will meticulously proof-read our final paper to weed out any grammatical errors.
>
>
> **Q4: The limitation that all classes of interest will occur in the unsupervised data allows the method to avoid forgetting.**
>
> **A:** As mentioned in L146-147 of our paper, our method does not require the unlabeled in-the-wild data to contain any objects from the base or novel classes. L147-149 further explains that our method is making use of the false positives from the in-the-wild data with non-overlapping object classes from the base and novel classes as training data.

---

> > ### Comment · Reviewer_m17V · 2021-08-16
> > **Thank you for your response!**
> >
> > I appreciate your clarifications.

---

### Official Review · Reviewer_DzAY · 2021-07-15

**Rating:** 5
**Confidence:** 3

**Summary:**

The paper proposes a technique to address the learning-without-forgetting problem when base and novel classes do not co-occur. The approach relies on sampling unlabeled in-the-wild data from another domain, to bridge base and novel classes. Results against a few recent baselines, as well as ablations, demonstrate the effectiveness of the approach.

**Limitations And Societal Impact:**

Not discussed

**Main Review:**

The proposed approach is reasonable, but other than the novel setting (non-co-occurrence) the key novelty evades me. There are also some descriptions that were unclear.

1. In particular, while it is clear why co-occurrence of base and novel classes shouldn't be assumed, it is not clear why strict non-co-occurrence is enforced (L135).

2. From the description in L152-166, it is not clear to me where the data to be used in D_{unlabel} comes from. The previous text talks about D_{base} and D_{novel}. This is also not shown (perhaps just not labeled) in Fig. 1.

3. The consistency check (L158) does not seem novel, and is in line with prior work on self-supervised learning (e.g. SimCLR).

4. The right-hand side of Fig. 2 needs more explanation (e.g. textual labels for the different components).

5. It was not clear to me why non-affection masks (where does the name come from?) alleviate confusion (L209).

6. Experiments are strong and convincing, including ablations.

Minor:
(a) Why is the subscript "s" used for novel classes, and "t" for base?
(b) Why not use D_{base}, D_{novel} as an argument to M_{stud}?
(c) Typos, D_{basel} (L153) and "are be" (L193)

**Time Spent Reviewing:**

1

---

> ### Author Response · Authors · 2021-08-10
> **Our response to Reviewer DzAY**
>
> **Q1: Other than the novel setting (non-co-occurrence), the key novelty evades me.**
>
> **A:** We disagree that our novelty only comes from the non co-occurrence setting. Our other key novelties are from our proposed approach in solving the non co-occurrence problem. Specifically, we propose the blind sampling strategy, and the class remodeling, image distillation with non-affection masks and the instance-level distillation components that eminently make up our dual-teacher framework for class incremental object detection. In other words, our major novelties and contributions come from both the problem setting and our proposed solution.
>
>
> **Q2: Why strict non-co-occurrence is enforced (L135)?**
>
> **A:** In practice, there is no guarantee on the co-occurrence of base and novel classes since the data can be obtained from significantly different settings. For example, the base classes from a bathroom setting where a base class of bathtub is unlikely to co-occur with novel classes from a bedroom setting. Thus, the co-occurrence assumption which existing works depend heavily on is impractical. We push the boundary by proposing an approach that can work under the extreme case where there is strictly no co-occurrence between the base and novel classes. We also show in our results that our method outperformed existing approaches when there is co-occurrence.
>
>
> **Q3: Where the data to be used in D_{unlabel} comes from (L152-166)?**
>
> **A:** As mentioned in L244-245 under the experimental setup section of our paper, we use MS COCO and Open Images datasets as D_{unlabel}. We deferred the mention of the exact datasets for D_{unlabel} to the experimental setup section to keep the descriptions of our approach general.
>
>
> **Q4: The consistency check is in line with prior work SimCLR (L158).**
>
> **A:** We believe that SimCLR is also not the first work to use the concept of consistency check. Nonetheless, it adopted the concept to learn visual representation and showed convincing results. In similar vein, to the best of our knowledge we are first to adopt consistency checks on our blind sampling of useful in-the-wild data and dual-teacher framework for class incremental object detection. Furthermore, we verified our approach with strong and convincing results.
>
>
> **Q5: The right-hand side of Fig. 2 needs more explanation.**
>
> **A:** We will include more text labels in Fig. 2 in our final paper as follows: [Figure 2](https://docs.google.com/presentation/d/19B_Ybu9gkq95rsQ6PWCRKkayDybuuPWB/edit?usp=sharing&ouid=104968873642613184581&rtpof=true&sd=true).
>
>
> **Q6: Why is it called “non-affection masks”? How does it alleviate confusion (L209)?**
>
> **A:** The non-affection masks are obtained from the pseudo ground truths generated from the blind sampling strategy. We call them non-affection masks because they are used to mask out negative instructions from the base (or novel) class teacher model that cause the student model to wrongfully suppress novel (or base) classes from the input images as background. In other words, we want the teacher/student models to have no effect (i.e. no affection) on the backgrounds that contain relevant foreground information.
>
>
> **Q7: Minor: (a) Why is the subscript "s" used for novel classes, and "t" for base? (b) Why not use D_{base}, D_{novel} as an argument to M_{stud}? (c) Typos.**
>
> **A:** (a) It was an arbitrary choice, we will modify them to “n” and “b” for clarity. (b) Information from D_{base} and D_{novel} are already encoded into the base and novel teacher models that are kept frozen during the dual-teacher distillation, and do not bring any substantial addition information compared to D_{unlabel} at this stage. (c) We will amend the typos.

---

> > ### Comment · Reviewer_DzAY · 2021-08-16
> > **thanks!**
> >
> > Thanks for the response!

---

### Official Review · Reviewer_zfwt · 2021-07-16

**Rating:** 6
**Confidence:** 3

**Summary:**

This paper investigates incremental object detection problem without making the assumption of base and novel classes occurring together, i.e. person (base) and TV(novel) objects being in the same image. The authors make the argument that this does not always happen in real-world scenarios, and they propose to sample in-the-wild data instead. They call this stage the "Blind sampling strategy", and this is achieved by applying the pre-trained Base and Novel models to the unlabeled data, and collecting those that have "positive" output. After the blind sampling strategy, a student model is learned by distilling the knowledge from the base and teacher models. Finally, the method is evaluated on PASCAL VOC and MS COCO datasets.

**Limitations And Societal Impact:**

The authors do not discuss limitation nor societal impacts in the submission.

**Main Review:**

The main contribution of this paper is to use the in-the-wild data, rather than the data with co-occurence for incremental object detection. The authors do not make the assumption that novel and base classes will co-occur in the images, which makes the problem more challenging and the solution more general. However, other than this, the technical contribution of this paper is very limited, as dual-teacher distillation is simply applied to distill the information between multiple models.

Regarding the blind sampling strategy, I would have found it beneficial if the authors had shown more ablations and experiments for this part since it is the main contribution. For example, what happens if the data is randomly selected rather than using pre-trained models? Can the pre-trained models minimize the impact of true negatives during the distillation in this case? Also what happens if different sources of unlabeled data are used? For example, does the performance change if Open Images dataset was used as the unlabeled data instead of MSCOCO for the PASCAL VOC experiments?

The dual-teacher distillation can be costly in terms of memory and complexity, as three models (two teachers and a student) need to be used during the training. Can the authors discuss why two teachers are required? For example, assuming that the novel data is available at the same time as the unlabeled data, can the student be learned with novel+unlabeled data while distilling information from a single teacher model (learned with base classes)?

Overall, the paper is nicely written and offers a more generalized solution for incremental object detection. It would benefit from additional discussions/experiments as described above. My initial evaluation is leaning towards acceptance.

**Time Spent Reviewing:**

1

---

> ### Author Response · Authors · 2021-08-10
> **Our response to Reviewer zfwt**
>
> **Q1: The technical contribution of this paper is very limited, as dual-teacher distillation is simply applied to distill the information between multiple models.**
>
> **A:** To the best of our knowledge, we are first to utilize the dual-teacher distillation framework on class incremental object detection. Furthermore, we meticulously design three novel components that allows the dual-teacher distillation framework to work on class incremental object detection: 1) A class remodeling step that remodels the irrelevant classes as background class in the base and novel model, respectively. This ensures the class probabilities of the disjoint set of classes in the base and novel models can be compared appropriately to the student model. 2) Image-level distillation with non-affection masks from the pseudo ground truth of the bounding boxes obtained in the blind sampling step. These non-affection masks are used to mask out the irrelevant regions of the feature maps in the distillation loss and eliminate the influence between base classes and novel classes. 3) Instance-level distillation with the response heatmaps of the object detectors, where the response heatmaps are essential for transferring both positive (high response regions) and negative (low response regions) information from the base and novel teacher models to the student model.
>
>
> **Q2: Ablations of blind sampling strategy.**
>
> **A:** Random selection from the in-the-wild data can lead to sub-optimal performance since there is no guarantee that the selected data contains positive information of the base and novel classes. We add the ablation for blinding sampling by a random selection of training data from the unlabeled dataset in Row 5 of [Table 7](https://docs.google.com/presentation/d/1i7SQBI18yBEAPCDNhhBrX-aHkhDpRkhO/edit?usp=sharing&ouid=104968873642613184581&rtpof=true&sd=true). We assign pseudo ground truth labels to these selected unlabeled data with the pre-trained models .  We can see that the performance under this setting is not as good as our proposed blind sampling strategy.
>
>
> **Q3: You should do experiments using the Open Images dataset as the unlabeled data instead of MSCOCO for the PASCAL VOC experiments.**
>
> **A:** We have reported the results in [Table 5](https://docs.google.com/presentation/d/12hqdXyZLON8TkGCShly_yhKioFbixGo-/edit?usp=sharing&ouid=104968873642613184581&rtpof=true&sd=true) of our paper, where the Open Images dataset is used as the unlabeled data instead of MSCOCO for the PASCAL VOC experiments. We can see that our approach still outperforms existing state-of-the-art under this setting.
>
>
> **Q4: The dual-teacher distillation can be costly in terms of memory and complexity. Why are two teacher models required?**
>
> **A:**  Although our approach can incur higher costs on the additional teacher model during training, our inference costs when only the student model is used are the same as existing single-teacher models. Nonetheless, these additional training costs are negligible since we can still comfortably train our model on a single 11GB 1080ti. Furthermore, our dual-teacher distillation ensures that unbiased knowledge from the base and novel teacher models are imparted to the student model, respectively.  Using a single teacher model is not ideal due to the confounding effect of the model parameters learned from the base classes when the novel+ unlabeled data are used.

---

### Official Review · Reviewer_SGgv · 2021-07-17

**Rating:** 6
**Confidence:** 3

**Summary:**

Authors address the problem of catastrophic forgetting in the context of object detection.
They propose a novel class-incremental object detection method based on Faster R-CNN for when there is no co-occurrence of base and novel object classes.
They also introduce a blind sampling strategy to select data from the large in-the-wild dataset for incremental learning.
Their dual-teacher distillation framework trained on this data outperforms SOTA on COCO and VOC datasets.


**Limitations And Societal Impact:**

Authors admit not discussing the limitations

(b) Did you describe the limitations of your work? [No]

**Main Review:**

Object detectors are an important class of models that are heavily used in practice and indeed one of the main challenges is adapting these models to learn to detect  new classes objects rather than training them from scratch. In this regard, it is a significant challenge that is addressed in this paper.


L 52: “we first introduce a blind sampling strategy to select relevant data from the in-the-wild data that contains large amounts of irrelevant images with neither the base nor novel class information.”
How do you ensure that? Do you mean the objects from the base and novel class do not exist in this dataset?!


L 69: Instead of assuming fully no co-occurrence of the base classes and the novel classes, wouldn’t it be more realistic to assume partial overlap as has been done in previous work?!




Evaluation

L 252: …  subsequently reduced by 0.1 after every 5 epochs? How? divided?


In table 1, It is odd that your method (last row) beats the baseline (row 1) where all data is seen at once? Why is that so?
Also, here the experiment is conducted in only one scenario (with tv being the novel class). It is important to repeat this experiment multiple times and also consider other permutations of base and novel classes to make sure results are statistically significant. Same argument applies to the results presented in other tables.


Results in Table 6 over COCO dataset does not seem complete and convincing! I think this is important as being the only result over COCO. Other experiments are over VOC which is easier to overfit. COCO is much more challenging and is used widely.


Typos:
L 307: Tables 4 shows -> tabe
L 363 Overall, our method can even outperform the previous state-of. (Move even to later in the sentence)






Authors have addressed an important and practical problem and results of their experiments are generally convincing although I have some concerns as raised above. I am leaning towards acceptance after considering the authors’ rebuttal.

**Time Spent Reviewing:**

3

---

> ### Author Response · Authors · 2021-08-10
> **Our response to Reviewer SGgv**
>
> **Q1: How to ensure the in-the-wild data contains neither the base nor novel classes information (L52)?**
>
> **A:** In practice, we cannot guarantee that the in-the-wild datasets do not contain any images from the base and novel classes. Furthermore, it should be noted that images in the in-the-wild dataset that contain the base and novel classes (i.e. co-occurrence) are inherently helpful in improving the performance of incremental learning; albeit these images might only exist in small quantities. To demonstrate the effectiveness of our proposed algorithm, we show results on the extreme case where there is no co-occurrence of the base and novel classes in the in-the-wild dataset. We preprocess the in-the-wild dataset by removing images that contain the base and novel classes using the ground truth object labels.
>
>
> **Q2: It is more realistic to assume partial overlap of the base and novel classes as has been done in previous work (L69).**
>
> **A:** We disagree that it is more realistic to assume partial overlap of the base and novel class datasets. This assumption always holds true in previous works because the base and novel class datasets are usually split from the same dataset under the same setting. However in a more realistic setting, images of the base and novel classes may be taken under different settings where co-occurrences are not possible. Our approach is designed to solve incremental learning under this extreme setting, and nonetheless still works in the presence of occurrences.
>
>
> **Q3: How to reduce the learning rate (L252)?**
>
> **A:** We reduce the initial learning rate by multiplying it by 0.1 after every 5 epochs.
>
>
> **Q4: Why does your method (last row, Table 1) beat the baseline (row 1, Table 1)?**
>
> **A:** The baseline in Row 1 is trained on training data without base and novel class co-occurrence to make fair comparison with our approach without co-occurrence ("w/o co-occur"). It can be seen from Row 6 that our approach without class overlap in the in-the-wild data (“w/o category) did not outperform the baseline in Row 1. In contrast, our method in the last row is trained on data with co-occurrence ("w co-occur") and with class overlap in the in-the-wild data ("w category") to make a fair comparison with [27] and [40]. We will indicate “w/o co-occur” in Row 1-3 for clarity in the final paper.
>
>
> **Q5: It is important to repeat this experiment multiple times and consider other permutations of base and novel classes.**
>
> **A:** We follow the 19+1 (“tv”) on the VOC test setting used in [27] and [40] for fair comparisons. Nonetheless, we also ran the experiments on various +1 novel class settings. The results are as follow: [Results of 19+1 on VOC test set](https://docs.google.com/presentation/d/1PrSQo3QyD0fxndW0bjjJQMwn_7FHILnU/edit?usp=sharing&ouid=104968873642613184581&rtpof=true&sd=true). We can see that our performance on various +1 novel class settings remains consistently good.
>
>
> **Q6: Results in Table 6 over the COCO dataset do not seem complete and convincing.**
>
> **A:** In Table 6, we follow [27] and [40] in reporting the results on the “40+40” COCO minival set. Unfortunately, we cannot show the “w/o co-occur” results from [27] and [40] due to a bug in [27] and no source code from [40].  Nonetheless, here we further supplement our results on COCO “w co-occur” for a more complete comparison with [27] and [40] under the “w co-occur” setting: [Table 6](https://docs.google.com/presentation/d/1ngFC0OsXXjxIADGE6gD-kqs-5SkTa7Yl/edit?usp=sharing&ouid=104968873642613184581&rtpof=true&sd=true). We can see that our method significantly outperforms [27] and [40] under the “w co-occur” setting.
>
>
> **Q7: Typos.**
>
> **A:** We will fix the typos in the final paper.

---

### Decision · Program_Chairs · 2021-09-27

**Decision:**

Accept (Poster)

**Comment:**

The paper proposes an approach for incremental object detection, which removes the need for co-occurrence of base and novel object classes in images in the training dataset. The reviewers scored the paper with (5, 6, 6, 6). After discussion and rebuttal,
they lean towards accepting the paper as a poster.